# Mathematical and Numerical Modeling of Repeated Floods from the Siret Basin, Romania, a Risk for Population, Environment, and Agriculture

Victorita Radulescu

Hydraulics, Hydraulic Machinery and Environmental Engineering Department, University Politehnica of Bucharest, Splaiul Independentei 313, 060042 Bucharest, Romania; vradul4@gmail.com; Tel.: +40-745-076-494

**Abstract:** In recent decades in Romania, no flood management plan has been implemented in natural riverbeds, although there are known areas that face repeated floods such as the Siret River basin. Practically every year, floods produce uncontrolled erosions and landslides in certain areas, followed by the deposition of sediments, usually on agricultural land, compromising crops indefinitely. This paper analyzes the natural transport capacity of the Siret River based on direct measurements and data recorded during the floods of 2005 and 2020. The mathematical model of the sediment transport is presented, starting with the upstream zone, from the confluence with its main tributaries: Bistrita, Trotus, and Cracau. The recorded flood hydrographs are used in this analysis to model the sediment transport for variable flow rates. The upstream flood hydrograph, the steady downstream level, and the initial riverbed cross-sections represent the boundary conditions. The mathematical model is numerically tested for the risk zones by determining the modifications of the riverbed cross-sections. The variation in time of the liquid and solid phases allows the estimation of the longitudinal riverbed shape with the floodable surfaces. To mitigate the effects of floods—and to protect the population, agricultural lands, and environment—some solutions are finally proposed.

**Keywords:** environmental engineering; floods; fluid flow measurements; hazardous areas; mathematical model; numerical model; sediment transport; turbulence

## 1. Introduction

In recent decades, global warming has led to climate changes with significant changes in hydrodynamic parameters, sometimes with abundant precipitation followed by floods and at other times, producing prolonged drought [1,2].

In many countries, floods still cause significant damage and even human casualties, flooding localities, destroying houses and roads, or covering agricultural land with banks rendering them practically unusable for several years [3,4]. Floods produce socio-economic effects and can even affect the population health of the respective areas by polluting the water sources, possible epidemics, etc. Only last year in three countries from Asia—India, Bangladesh, and Nepal—17.5 million inhabitants were affected by floods and threatened by various diseases [5]. In 2007 more than 46 million people in India, Nepal, Bangladesh, and Pakistan were also affected by torrential rains and floods, [6].

Europe also faces the same problems. Last year in Spain and in 2021 in the U.K. and six other countries, including Austria, Belgium, Croatia, Germany, Italy, Luxembourg, the Netherlands, and Switzerland, floods caused significant damages and even human losses [7,8]. Eastern Europe, where investments in flood management and avoidance of their effects were reduced in the last decades, is much more affected [9,10].

Romania has many zones, such as the Siret river basin, Crisuri basin, and Tarnavelor basin, that face floods every year; sometimes even more in the same year for the same area, as was the case for Siret in 2020. Romania has an average hydrographic basin on a European scale, with natural riverbeds prone to flooding [11,12].

This paper analyzes the watershed of the Siret River, which, with a length of 559 km, is the largest and most vulnerable in the country to repeated floods. It receives almost all watercourses from the northeastern part of the country until it flows into the Danube River. Because many of its tributaries have natural riverbeds, flooding is recorded in years with significant precipitation [13,14]. The flow in this watershed increases significantly from April, with maxima in May and June [15]. Many years recorded repeated floods, for example, 1999 with floods in May and June, respectively, and 2005 with floods in June and July. In 2020, however, there were three floods in less than two months [16]. Many localities near these watercourses are subject to flooding almost every year.

Many areas have been designated protected zones because they are on the routes of migratory birds to the Danube Delta or warmer countries [17,18]. As a member of the EU, Romania has signed many documents related to environmental protection and habitat [19,20], including of natural riverbeds [21], and imposing the maximum allowed quantities of pollutants discharged into these watercourses. There are many nature reserves in this area with protected species of plants, birds, and animals, as well as sites protected by the EU Council Directive 2009/147/EC, especially in the Bistrita River area [22–24].

Rivers Trotus, Cracau, Bistrita, Putna, and tributaries of the Siret, are also frequently flooded. Thus, there were two floods in June and July 2005 that covered thousands of hectares of agricultural land, four floods in 2015, and three major floods in about two months in 2020. The last ones had high flow rates, produced material damage, covered agricultural lands and flooded many localities. Improperly managed floods cause significant damage, destruction of bridges and roads, and affect protected areas and environment, agricultural land and houses, and in many cases, can produce even human losses [25,26].

If the floods occur again, the damage will be even more visible. In 2 counties, 182 localities were affected in 2020, with 5467 houses; 26 schools; 16 kindergartens; 10 hospitals; and more than 170,000 ha of agricultural land covered by water. Moreover, 1330 km of roads were damaged by water and landslides, 1024 small and medium bridges were destroyed, as were 6 km of railways [18]. Floods transport sediments, which can cause bank erosion and landslides downstream, typically covering agricultural land and roads. Floods must be managed correctly and at the proper time, to avoid these ecological and economic problems.

In this paper, the Siret River basin is analyzed, an area frequently faced with floods. As a case study, the riverbed modifications after the three floods recorded in 2020 are modeled. The first one was recorded on 13–14 May, the second on 24–25 June, and the third on 20–21 July, approximately nine weeks after the first flood. Their cumulative effects are analyzed, illustrating the flooded surfaces presented comparatively.

Considering that this area is often affected by floods, the mathematical and numerical modeling of the specific areas affected by floods is beneficial to ensure the protection of the population. The modeling is carried out in concrete conditions, with data calibrated with direct measurements in the field or from the hydrographic history of the area. The flood hydrograph, similar to those recorded in 2020, and the initial cross-sections are considered known. A mathematical and numerical model of sediment transport is realized and modeled the flood. The final form of the cross-sections will be determined after the flood [27,28]. Places where erosion occurs or where sedimentary material is deposited will be highlighted, associated with the zones where the minor riverbed cannot transport the tributary flows brought by the flood. The dimension, type, and nature of the sediments are selected based on field measurements [29,30]. Water parameters are also known, as are flow rates recorded in several monitoring stations and some environmental data, known by their influence on the hydrological regime [31,32]. Based on obtained data from numerical modeling and calibrated on the measured values, the longitudinal profile of the riverbed is estimated [33,34]. Since a large distance is covered, it is divided into six zones in which the calibration parameters are modified, such as the slope of the river, the nature and type of sediments, the flow rates as more tributaries are encountered, etc. The modifications in the transverse profiles and of the longitudinal riverbed can undergo rapid evolution, affecting

the hydrological regime in the entire area [35,36]. Numerical modeling is carried out on cross-sections, ensuring the mass balance of flows and sediments between the entry and exit sections of each domain. By continuity of the flow, numerical modeling determines the final shape and dimensions of the riverbed in cross-sections and its longitudinal profile [37,38]. The maximum allowable flow rates are determined with certain probabilities. Areas likely to be flooded are illustrated. Based on the obtained results by numerical modeling, for the risk areas that must be rehabilitated a management plan is proposed for the entire hydrographic basin, for conservation environment, agricultural lands, and natural habitats.

## 2. Materials and Methods

The hydrographic basin of the Siret River has a catchments area of 42,980 $km^2$, accounting for 18% of the country's land. The average flow rate is Q = 225 $m^3/s$, with a volume of water $V_w$ = 7100 × $10^6$ $m^3$, and an annual stock of $Q_s$ = 0.9 $km^3/year$.

The hydrologic activity was carried out in accordance with a program of observations and measurements in the next stations:

- 79 hydrometric stations; 73 with daily transmissions; 60 automatic stations;
- 110 rain gauges with daily communication;
- 6 evaporimeter stations;
- 256 hydro-geological boreholes, 80 of which are equipped with automatic stations;
- 356 cross-sections, 5 of them are chosen for presentation, as being representative;
- 71 major consumers were tracked systematically.

For numerical modeling, some parameters were monitored:

- Tributary flow rates and flood hydrographs with a specific frequency in time;
- Hydrological data from hydrometric stations for the last decades;
- Precipitation, evaporation, and water consumption for the analyzed basin;
- Environmental parameters such as temperature, wind speed and direction, solar radiation, pressure, and so on.
- Sediments: classification, dimensions, structure, and distribution.

### 2.1. Local Conditions

The Siret River has the largest hydrographic basin in the country with the highest collected flow rates, as shown in Figure 1a in purple. With a continental climate, it has certain main characteristics:

- Average annual temperature: $T_{av}$ = 8.5 °C;
- Average minimum temperature, in January $T_{min}$ = −4.5 °C; average maximum temperature (in July in the south and August in the north) $T_{max}$ = 20 °C;
- Relative humidity reaches 80% for approximately 6–8 months per year;
- Average wind speed of 3–6 m/s at 10 m from the ground;
- Annual precipitation averages 500–900 mm, with a wide range from July to November; precipitation is higher here than elsewhere in the country, reaching 450 mm even during the warm seasons.

Three numbers are mentioned in Figure 1b, where the Siret hydrographic basin is highlighted in green. Based on data collected over the last two decades, the first picture illustrates the floodable surface in thousands of hectares and the second, the population exposed to floods in thousands of inhabitants.

The rivers Bistrita (27%), Trotus (10%), Moldova (12%), and Suceava (12%) are among its main tributaries with the greatest water catchment. Although the Cracau River has a lower average flow rate (about 10%), it becomes significant in years with heavy precipitation, contributing significantly to flood propagation. All of these rivers are vulnerable to flooding. Bistriţa is the largest tributary, with an average flow rate of Q= 55 $m^3/s$, a length of L = 283 km, an elevation difference of h = 372 m, and an area of A = 7039 $km^2$ with 193 tributaries. Table 1 shows some measurements realized in 5 Bistrita River monitoring

stations during 2019, considered an average year. Values were obtained from the Romanian National Authority Apele Romane, from its report for 2020 [39].

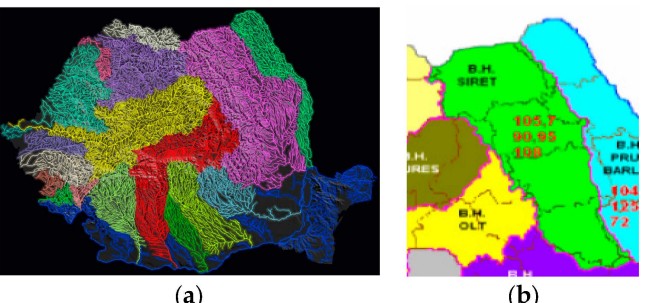

|            | (**a**) |            |            | (**b**) |            |

**Figure 1.** The hydrographic basin of the Siret River: (**a**) nationally occupied surface; (**b**) data recorded in the last decades.

**Table 1.** Average flow rates in some monitoring stations.

| Sections/Q[m$^3$/s] | SH Carlibaba | SH Carnu | SH Brosteni | SH Vaduri | SH Bacau |
|---|---|---|---|---|---|
| X | 27.96 | 33.25 | 33.52 | 33.85 | 37.99 |
| XI | 26.42 | 30.99 | 31.23 | 31.50 | 40.40 |
| XII | 21.48 | 25.36 | 25.57 | 25.81 | 33.83 |
| I | 17.96 | 21.25 | 21.38 | 21.59 | 28.49 |
| II | 18.13 | 21.46 | 21.63 | 21.85 | 27.05 |
| III | 32.11 | 38.41 | 38.77 | 39.19 | 44.61 |
| IV | 80.30 | 97.35 | 98.11 | 99.28 | 107.17 |
| V | 92.21 | 109.53 | 110.47 | 111.55 | 118.18 |
| VI | 73.15 | 87.65 | 88.41 | 89.34 | 96.22 |
| VII | 59.37 | 70.91 | 71.49 | 72.22 | 76.38 |
| VIII | 43.96 | 52.56 | 53.02 | 53.56 | 57.15 |
| IX | 33.10 | 39.51 | 39.85 | 40.29 | 41.87 |
| Annual Average | 43.85 | 52.35 | 52.79 | 53.34 | 59.11 |

The case study was based on measurements realized in more than 60 measuring points along the Cracău, Bistriţa, Trotus, and Siret rivers from 2006 to 2019, by a research team including specialists from different domains and licensed and master's students from the University Politehnica of Bucharest. Flow characteristics for the sector of interest were updated with data collected after the 2020 floods. The high moisture content of the mountain soils ensures almost continuous infiltration of groundwater into rivers. The soil in the central and southern areas is alluvial, consisting primarily of sand and clays. Morphological and hydro-chemical data show that sediments were transported more than 245 km during floods, like those in 2005 and 2015. Due to the presence of chernoziom, faeoziom, and preluvos, the soil in the northern area has a high capacity for infiltration and water retention.

Heavy, torrential snow and rains were recorded between 1 January and 10 May 2005, with totals ranging from 100 to 400 mm. Because the water in soil was still frozen, especially in the mountains, it was practically impermeable. As a result of the increased humidity and the prolonged frost, 70–80% of the precipitation fell directly into the drainage system, producing catastrophic floods (Figure 2a–c). The pictures were made with drones in 2005 and 2020, by "Romanian Waters", and reported to the local authorities [40]. This included the entire Bistriţa hydrographic basin and extended downstream to Siret. Unfortunately, all the downstream lakes and riverbeds were already full, so the flood covered almost the entire length of the Siret course.

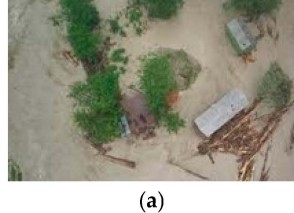
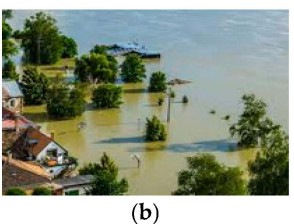
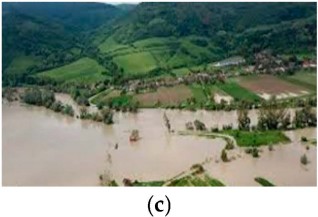

(**a**)        (**b**)        (**c**)

**Figure 2.** Floods on Bistrita River: (**a**) 2005, near Batca Doamnei; (**b**) 2005, near Prelunca; (**c**) 2020, near Piatra Neamt.

Even though the Siret River has a vast hydrographic basin, it frequently floods. During normal conditions, the riverbed has a width of 70–100 m and depths of up to h = 6–8 m, but during floods, the width increases to more than 300 m and depths of h = 15–20 m, flooding large agricultural surfaces and localities, as shown in Figure 3a–d. The pictures were realized with drones by the National Authority "Romanian Waters" and transmitted to the local authorities. Because of the large amount of sediment transported, the riverbed changes very quickly, as evidenced by its meanders. Figure 3b illustrates the riverbed axis at the beginning of 2020, as seen from a drone, and at the end of the year and after the passage of the three floods.

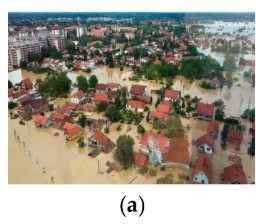
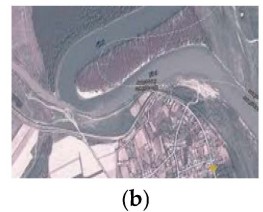
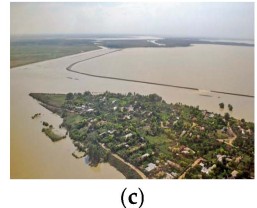
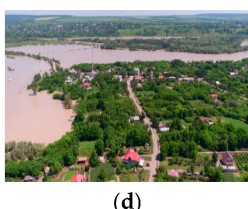

(**a**)        (**b**)        (**c**)        (**d**)

**Figure 3.** Floods on Siret River: (**a**) 2005, near Piatra Neamt; (**b**) 2020, after the third flood, near Ramnicu Sarat; (**c**) 2015, near Bacau; (**d**) 2020, near Focsani.

Table 2 lists the parameters used in the numerical modeling, where F (km$^2$)—the surface from which data is collected; $Q_{av}$ (m$^3$/s), $Q_{max}$ (m$^3$/s) is the average and maximum flow rate; R (kg/s)—the volume of transported alluvium determined by direct measurements for the mentioned location; and T ($^\circ$C) is the water temperature.

**Table 2.** Main parameters considered for modeling, registered in 2019.

| | | | | Hydrologic Parameters | | | | Water Parameters | | |
|---|---|---|---|---|---|---|---|---|---|---|
| Nr. | River | Station | F (km$^2$) | $Q_{av}$ (m$^3$/s) | $Q_{max}$ (m$^3$/s) | R (kg/s) | Date | T ($^\circ$C) | pH | Organic Mg KmnO$_4$/L |
| 1 | Siret | Lespezi | 58,744 | 36.6 | 1825 | 67.8 | 04.03 | 6.1 | 7.5 | 10.97 |
| 2 | Siret | Dragesti | 11,811 | 76.8 | 2650 | 126 | 06.05 | 6.4 | 7.5 | 12.09 |
| 3 | Siret | Lungoci | 36,030 | 208 | 3950 | 349 | 06.05 | 6.8 | 7.6 | 14.16 |
| 4 | Suceava | Itcani | 2330 | 16.4 | 1725 | 15.1 | 11.08 | 14.0 | 7.2 | 9.1 |
| 5 | Bistrita | Roman | 4285 | 32 | 1925 | 40.1 | 11.08 | 15.0 | 7.3 | 9.4 |
| 6 | Bistrita | Frumosu | 2816 | 37.5 | 1320 | 7.42 | 11.08 | 15.5 | 7.3 | 9.7 |
| 7 | Trotus | Vranceni | 4077 | 35.4 | 2500 | 37.8 | 12.09 | 14.8 | 7.4 | 6.73 |
| 8 | Putna | Botariu | 2518 | 16.5 | 1790 | 87.2 | 12.09 | 13.4 | 7.3 | 8.1 |

Table 3 illustrates the sediment structure on the Siret River in the same sections. Table 4 depicts the maximum flow rates recorded on the Siret River over the last decades. The shape of the flood hydrograph is determined by the rise and fall zones, bounded in the upper part by Qmax. Covered surfaces by water, the reached level, the amount of sediment transported, and finally, the damages produced are the flood's distinguishing characteristics.

In the last decades, 14 catastrophic floods in the Siret basin have been recorded, with human losses, dead animals, destroyed houses, flooded roads and agricultural lands, and destroyed bridges. The average precipitation from 2007 fitted the characteristics of a dry year. Moreover, 4 months—from May to August—were the driest years recorded in the last 40 years. For the first time in Romania, Codes Orange and Red for high temperatures in July and August were recorded.

**Table 3.** Structure of sediments in three sections on Siret River.

| Nr. | Parameter | SH Lespezi | SH Dragesti | SH Lungoci |
|---|---|---|---|---|
| 1 | Humidity (105 °C) % | 55.46 | 53.38 | 55.66 |
| 2 | Organic substances (%) | 6.95 | 5.93 | 8.36 |
| 3 | Mineral substances (%) | 93.05 | 94.07 | 91.64 |
| 4 | $NH^+_4$ (mg/100 g) | 5.85 | 7.27 | 8.01 |
| 5 | $NO^-_3$ (mg/100 g) | 0.33 | 0.41 | 0.19 |
| 6 | $PO^{3-}_4$ (mg/100 g) | 0.068 | 0.072 | 0.086 |
| 7 | N- $NH^+_4$ +N- $NO^-_3$ (mg/100 g) | 4.61 | 5.74 | 6.27 |
| 8 | P- $PO^{3-}_4$ (mg/100 g) | 0.022 | 0.023 | 0.028 |
| 10 | $N_{included}/P_{included}$ | 209.54 | 249.56 | 223.93 |

**Table 4.** Maximum flow rates recorded in Station Lungoci.

| Year | 1989 | 1990 | 1991 | 1992 | 1993 | 1994 | 1995 | 1996 | 1997 | 1998 | 1999 | 2000 | 2001 | 2002 | 2003 | 2004 |
|---|---|---|---|---|---|---|---|---|---|---|---|---|---|---|---|---|
| $Q_{max}$ | 1535 | 1260 | 1860 | 889 | 1320 | 2460 | 1400 | 334 | 277 | 2620 | 1370 | 275 | 3270 | 2045 | 1020 | 603 |
| 2005 | 2006 | 2007 | 2008 | 2009 | 2010 | 2011 | 2012 | 2013 | 2014 | 2015 | 2016 | 2017 | 2018 | 2019 | 2020 | 2021 |
| 1614 | 1120 | 1040 | 1380 | 830 | 447 | 435 | 2200 | 796 | 727 | 4650 | 1375 | 785 | 2068 | 3024 | 2022 | 4653 |

The years before 2010 had similar structures to dry years, so massive sediment deposits appeared in the wider areas of the river cross-sections, where the flow rates are lower.

In these conditions, after the snow melted associated with the recorded abundant precipitation riverbed was clogged in 2005. High flow rates appeared on Bistrita, Cracau, and Siret rivers, followed by massive floods. The same situation was recorded in 2015. Floods were recorded and repeated whenever the rains were heavier. However, the most severe, repetitive floods were recorded in 2020 two years after when the flow rates of the Siret tributaries exceeded the multi-year average.

*2.2. Mathematical Model of the Sediment Transport*

Flood risk analysis focuses on determining the transverse and longitudinal profiles for the analyzed riverbed sector, assuming they are continuously changing [41,42]. The sediment transport and tributary flow rates represent a major problem in watercourse management. When the riverbed is changed due to natural causes—erosion or human interference—excavations, a transitory phase of degradation, are initiated. This is not always advantageous in the case of flood management. The analysis of the transport capacity of the Siret River is carried out in five transversal sections known to be prone to flooding, which delimit the sections in which the longitudinal riverbed is divided. The mathematical model of sediment transport is based on the mass balance from one cross-section to another. The exit section from one sector becomes the entrance to the next one. The risk areas are determined from places where lateral discharges occur and by imposing the condition of flow continuity. The transport of sediments (erosion or deposition) along the riverbed is based on measurements made in the field regarding the type, nature, and size of the sediments in each sector. Thus, the longitudinal profile of the Siret River is determined step by step. The filler material is assumed to have the greatest influence on the final riverbed balance. The contribution of the suspended material is neglected, assuming that it is transported downstream. The non-uniformity of the vertical size distribution is neglected, by assuming that sediments with a larger size are reduced in percentage in

the erosion process. If the thickness of the alluvial bed is less than or equal to a certain length of influence, these assumptions can be accepted [43]. The Ackers and White method is used [44,45]. They consider how coarse sediment movement is caused by shear stress on the bed, while fine sediments are transported in suspension. The method allows the determination of erosions and the transport of sediments from the alluvial bed during a flood.

For the sediments, four diameters are considered for each analyzed sector. Based on this, an equivalent diameter is calculated for each cross-section, where: $d[m]$—average diameter of sediments, $g[m/s^2]$—gravitational acceleration. $v[m^2/s]$—kinematic viscosity of water, $\gamma_s$, $\gamma$—sediment and water specific weight:

$$d_* = d\left[\frac{g}{v^2}\left(\frac{\gamma_s}{\gamma} - 1\right)\right]^{1/3}, \tag{1}$$

The sediment mobility is determined by:

$$F_g = U_*^{C_1}\left[gd\left(\frac{\gamma_s}{\gamma} - 1\right)^{-1/2}\right] \cdot \left[\frac{V}{\sqrt{32}\log(10h/d)}\right]^{1-C1}, \tag{2}$$

where: $U_*$—the shear velocity, $U_* = \sqrt{\tau_0/\rho}$, $\tau_0$—shear stress in riverbed, $V$-average velocity, $h$-depth of water.

For: $d_* > 60$, the coefficients are: $C_1 = 0.0$; $C_2 = 0.025$, $C_3 = 0.17$, $C_4 = 1.5$ and for $1 < d_* < 60$, $C_1 = 1 - 0.56\log d_*$; $C_2 = e^{2.86\log d_* - 3.53 - (\log d_*)2}$; $C_3 = 0.23/d_*^{1/2} + 0.14$; $C_4 = 9.66/d_* + 1.34$.

The dimensionless sediment transport function $G_{gr}$ and X-the sediments concentration in ppm (parts per million) are:

$$G_{gr} = C_2\left(\frac{F_g}{C_3} - 1\right)^{C_4}, \quad X = G_p - \frac{d}{h}\frac{\gamma_s}{\gamma}\left(\frac{V}{U_*}\right)^{C1}, \tag{3}$$

Finally, the total load of sediments may be determined as $Q_{T = X \cdot Q}$, where $Q(m^3/s)$ is the flow rate of the river.

### 2.3. Mathematical Model of the River Flow

As boundary and initial conditions, the flow variation is analyzed through the flood hydrographs, Figure 4a–c and direct measurements of sediment structure. At the start, the initial conditions for the main variables must be specified for the entire domain.

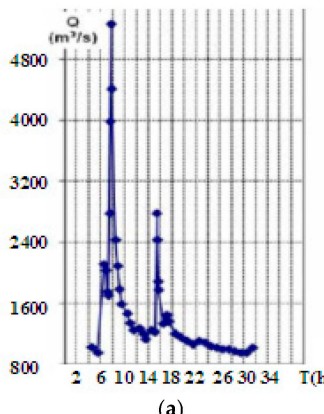
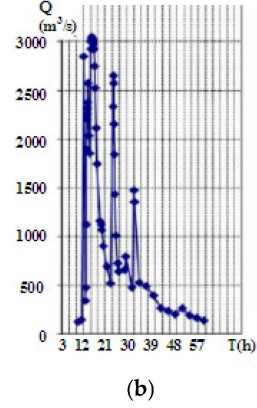
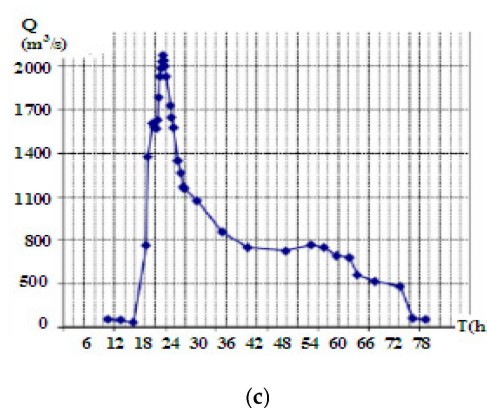

(a)  (b)  (c)

**Figure 4.** Floods on Siret River in 2020: (**a**) May 13–14; (**b**) June 24–25; (**c**) July 20–21.

The longitudinal profile is structured in four sectors with different characteristics, sediments structure and dispersion, and variable flow rates (depending on tributaries).

The exit from one sector becomes the entry into the next one, ensuring flow and sediment transport continuity. The input of suspended material is neglected, assuming high flow rates of transport. The proposed mathematical model simulates the effects of longitudinal transport and lateral discharges in case of floods [44,45].

The selected area has length-L; slope-$S_0$; width-B; A-cross-section; H-water level; $Q$—flow rate; and $Q = f(T)$ the flood hydrograph, where T-time. L is thought to be long enough to allow accurate determination of alluvial riverbed changes. As discussed in Section 2.2, the volume rate of transported sediments is $Q_T/g_s$, where $Q_T$-is the flow rate of water with sediments. In section 1 (S1-Entrance) it is: $Q_t - \frac{\partial Q_T}{\partial x}\frac{dx}{2}$ and in section 2 (S2-Exit): $Q_t + \frac{\partial Q_T}{\partial x}\frac{dx}{2}$, where x is the dimension in longitudinal direction. The net amount of sediments that change the river bed between section 1 and section 2 is: $\frac{\partial}{\partial t}[AHdx(1-\lambda)]$, where λ—porosity in bed. Combining these relations for unsteady flows, the continuity equation of sediment quantity conservation became:

$$\frac{\partial H}{\partial t} + \frac{1}{B\gamma_s(1-\lambda)}\frac{\partial Q_T}{\partial x} = 0, \tag{4}$$

If the thickness of sediments is small, the flow rate increases. The alluvial bed does not change in time, if the flow rate is constant. To solve this equation, the equations of motion and the continuity equation for the tributary flows are required [46].

### 2.4. Mathematical Model of Fluid Flow and Continuity of the Sediment Transport

Equation of continuity in riverbed may be written:

$$B\frac{\partial H}{\partial t} + \frac{\partial Q_T}{\partial x} = q_l, \tag{5}$$

where: $q_l$—the flow rate from the main current, $t$-time, $S_f$—the friction slope, determined from the Manning formula for rough surfaces, $S_f = V^2 n^2 / R^{4/3}$.

Equation of flow motion may be written:

$$\frac{\partial V}{\partial t} + g\frac{\partial}{\partial x}\left(\frac{V^2}{2g} + h\right) = g\left(S_0 - S_f\right), \tag{6}$$

Applying the Leibniz's rule and integrating, the final weight $G_f$ of sediments is:

$$G_f = G_i - C_g = G_i - C' \cdot G'_f 0 \leq C' \leq 1, \tag{7}$$

where: $C_g$—weight of sediments absorbed in the mainstream during a time interval Δt and transported downstream in the cross-section $S_2$, $G_f$—the weight of sediments which further, in the next sector may be transported or deposited, $G_i$—the total weight of the entered sediments in the cross-section $S_1$, and $G'_f$—only the transported sediments. The constant $C' = 0.043$ for the first two sectors, $C' = 0.06$ for the third, and $C' = 0.084$ for the last one, selected based on the sediment characteristics for each zone.

Following the Toffaletti scheme, with $t_1^{(i)}$ and $t_2^{(i)}$ the weights of diameters $d_i$ fractions in the solid discharge, the sediments removed from the bed is $C'\left(t_1^{(i)} - t_2^{(i)}\right)/2$, while $b_1^{(i)}G_f = b_2^{(i)}\left(G_i - C_g\right)$ represents the weight of fractions remained in bed; $b^{(i)}$ is the weight of fraction "$i$" in riverbed [47].

The weight balance in the alluvial riverbed for the fraction "$i$" becomes at the end of Δt interval, with $Q_1$, $Q_2$—the inflow and outflow discharge:

$$b_2^{(i)} = b_1^{(i)}\left(1 + C'\right) - C'\left(t_1^{(i)} + t_2(i)\right)/2, \tag{8}$$

The total weight of sediments in the liquid phase can be calculated by estimating the solid discharge $Q_l(x)$ that passes through the cross-section in a time interval equal to dx/V [27]. The weigh balance of the liquid became:

$$\frac{\Delta t\left(Q_{t1} \cdot t_1^{(i)} + Q_{t2} \cdot t_2^{(i)}\right)}{2} = \frac{LQ_{t1}}{2v} - \frac{LQ_{t2}}{2v} + C \cdot B \cdot L\frac{t_1^{(i)} + t_2^{(i)}}{2}, \tag{9}$$

The final weights of sediments in the mainstream are known at a time interval $\Delta t$ by summing both terms of equation for all "$i$" fractions. An iterative system is obtained by the sequential changes in the riverbed structure of sediments, for unsteady flow rate. Finally, the erosion thickness during $\Delta t$ is:

$$\Delta H = 2\frac{G_1 - G_2}{(\gamma_{s1} + \gamma_{s2})}, \tag{10}$$

The Saint-Vénant system consists of two nonlinear partial differential equations of the hyperbolic type, with the flow discharge $Q$ and water level H as unknown functions and the $x$ and $t$-time as independent variables. A computational grid is defined to solve this system. The mathematical model starts with the riverbed profile cross-sections, environmental conditions, distances between sections, geographic variation of the slope, riverbed roughness along the entire analyzed distance, maximum flow rate, time interval, sediment concentration, and flood hydrograph.

### 2.5. Numerical Modeling of Floods

Initial data for numerical modeling are obtained through direct measurements and bathymetric data for transverse and longitudinal profiles, flood hydrographs, and water levels in cross-sections at different flow rates of tributaries. The flood-prone areas have been identified. The main steps in numerical modeling are:

- Selection of the interest area with significant cross-sections on the longitudinal profile, Figure 5a–d. Figure 5c,d represents two cross-sections from the selected area, at a low flow-rate of water, for a selected time step;
- Selection of structures with direct impact on the water flow for these sectors, such as spillways, dams, weirs, bridges, and decks, required as initial conditions;
- Granulometry, type, and nature of sediments from the minor and major riverbeds in the selected sections, to assign appropriate values for sediment transport and floods.

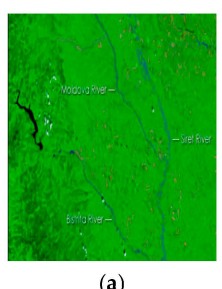 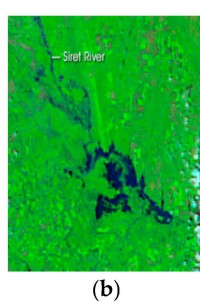 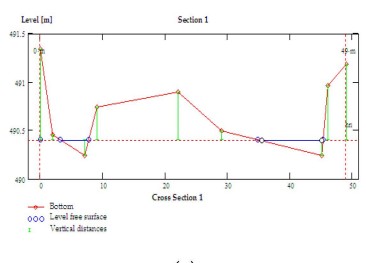 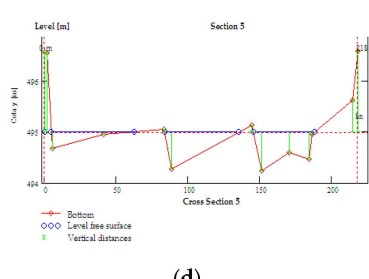

(**a**) (**b**) (**c**) (**d**)

**Figure 5.** Area of interest: (**a**) Longitudinal profile of tributaries; (**b**) Longitudinal profile Siret; (**c**), (**d**) Cross-sections.

The model was initially tested for tributaries flow rates, to compare the results with data known from direct measurements. This is a necessary step to verify the correct calibration of the model. The Digital Terrain Model (DTM) was calibrated by introducing the boundary conditions, such as hilly areas or protection dykes from some localities, but also the low areas of the major riverbed or the agricultural lands without protection.

The discretization cells are of 50 m, with the hydraulic parameters covering the entire area of analysis, for flow rates considered from tributaries ranging from 500 to 1200 m³/s

and reaching up to 4600 m$^3$/s for floods on Siret. Some rules must be respected when schematizing the numerical model:

- Schematization methodology is chosen so the level of accuracy remains the same, with good precision both at low and high flow rates, when the cross-section is partially or full of water and overflows in the major riverbed, localities, or agricultural lands;
- The cross-sections in the minor riverbed are chosen so the dams that are above the elevations in the major riverbed are represented as correctly as possible;
- Different average roughness values are used in the minor and major riverbed starting from the upstream to downstream sector, considering the composition of the alluvial bed, potential vegetation islands, and sinuosity of the water course. For better precision, the roughness was determined using the values obtained by measurements with non-homogeneous Manning values in the minor bed of 0.018–0.025 s/m$^{1/3}$, the major bed of 0.032–0.05 s/m$^{1/3}$, and flooded localities of up to 0.08–1 s/m$^{1/3}$;
- For the input upstream limit condition, the second recorded flood hydrograph from 2020 was used, as being the worst case. The level of the reservoir near Bacau city was considered the output, the downstream border.

The model was initially tested for permanent and small flow rates of 400–600 m$^3$/s before being run for the flood mode. These tests were carried out to validate the model, the selected sections and roughness in the cross-sections. It was assured the flow continuity for the selected river sector, considering the main tributaries and sediments characteristics. The differences obtained compared to direct measurements in the field were approximately 40–60 cm, indicating that the model was calibrated.

The flooded areas and risk factors were assessed in two scenarios. The first one is a flood coming from Bistrita River, a major tributary. The second assumes a smaller but longer flood hydrograph. The maps for flood risk were created and compared to satellite or drone images.

Assuming a constant space-step $\Delta x$, the Equations (7) and (10) may be approximated:

$$
\begin{aligned}
&\frac{1}{4\Delta x}L_j^{(m)}\left(h_{j+1}^{m+1}-h_{j-1}^{m+1}+h_{j+1}^{m}\right)+\frac{1}{\Delta t}\left(Q_j^{m+1}-Q_j^{m}\right)+\frac{qQ_j^{m+1}\left|Q_j^{m}\right|}{(C^2AR)^m}-\\
&\qquad\frac{1}{2\Delta t}M_j^{(m)}\left(h_{j+1}^{m+1}-h_{j-1}^{m+1}+h_{j+1}^{m}-h_{j-1}^{m}\right)+\\
&\qquad\left(\tfrac{\alpha q}{A}\right)_j^{(m)}\cdot\left(Q_j^{m+1}+Q_j^{m}\right)-\left(\tfrac{\alpha Q^2}{A^2}\right)_j^{(m)}\cdot\frac{A_{j+1}^{(m+1)a}\cdot A_{j-1}^{(m)a}}{\Delta t}=0
\end{aligned}
\tag{11}
$$

$$
\frac{1}{4\Delta x}\left(Q_{j+1}^{m+1}-Q_{j-1}^{m+1}+Q_{j+1}^{m}-Q_{j-1}^{m1}\right)+B_j^{(m)}\frac{h_j^{m+1}-h_j^{m}}{\Delta t}-q_j^{(m)}=0
\tag{12}
$$

where $L = gA - \frac{\alpha Q^2}{A^2}B$ and $M = \frac{2\alpha Q}{A}B$. This notation specifies the discharge $Q$ value at time m·$\Delta$t for grid point $x_j$. The superscript $(m)$ represents the time moment $(m+1)/2\cdot\Delta t$, where $A_{j+1}^{(m)a}$ represents the cross-section area for the grid point $x_{j+1}$, which is considered, at this time, an average water elevation between two adjacent h-grid points values. To avoid computational instability, the terms are expressed as a combination of known values $Q^m$ and unknown terms $Q^{m+1}$. Relations (11) and (12) become:

$$
\begin{aligned}
A_j^1 Q_{j+1}^{m+1}+B_j^1 Q_j^{m+1}+C_j^1 Q_{j-1}^{m+1}=D_j^1\\
A_j^2 h_{j+1}^{m+1}+B_j^2 h_j^{m+1}+C_j^2 h_{j-1}^{m+1}=D_j^2
\end{aligned}
\tag{13}
$$

where: $A_j^1 = \frac{1}{4\Delta x}$; $A_j^2 = A_j^1 L_1^{(m)} - \frac{M_j^{(m)}}{2\Delta t}$;

$B_j^1 = \frac{B^{(m)}}{\Delta t} = \frac{1}{\Delta t}+\frac{q\left|Q_j^m\right|}{(C^2AR)_j^{(m)}}$; $B_{j-1}^2 = \frac{-C_j^1}{A_j^1 B_j^1+B_j^1}$; $B_{j-1}^1 = \frac{-C_j^2}{A_j^2 B_j^2+B_j^2}$;

$C_j^1 = A_j^1$; $C_j^2 = C_j^2 L_j^{(m)}\frac{M_j^{(m)}}{2\Delta t}$;

$$D_j^1 = A_j^1 Q_{j-1}^m + B_j^1 h_j^m + C_j^1 Q_{j+1}^m + q_j^{(m)}; \text{ and}$$

$$D_j^2 = A_j^1 h_{j-1}^m + \frac{1}{\Delta t} - \left(\frac{\alpha q}{A}\right)_j^{(m)} Q_j^m + C_j^2 h_{j+1}^m + \frac{1}{\Delta x}\left(\frac{\alpha Q^2}{A^2}\right)_j^{(m)}\left(A_{j+1}^{(m)a} - A_{j-11}^{(m)a}\right)$$

The coefficients are calculated for the $h$ and $Q$ grid points, with the exception of the terms with $a$-superscript, which are calculated for the $h$-points with $h_j^a = \frac{h_{j+1}^a + h_{j-1}^a}{2}$. The system is solved by two simultaneous tri-diagonal double sweep algorithms having $Q_{j+1}^{m+1}$ and $h_j^{m+1}$ associated with $Q_j^{m+1}$ and $h_{j+1}^{m+1}$, interrelated by the next equations:

$$Q_j^{m+1} = E_j^1 \cdot h_j^{m+1} \cdot F_j^1 \tag{14}$$

$$h_{j+1}^{m+1} = E_j^2 \cdot h_j^{m+1} \cdot F_j^2 \tag{15}$$

where:

$$E_{j-1}^1 = \frac{-C_j^2}{A_j^2 E_j^2 + B_j^2}; \ E_{j-1}^2 = \frac{-C_j^1}{A_j^1 E_j^1 + B_j^1};$$

$$F_{j-1}^1 = \frac{D_j^2 - A_j^2}{A_j^2 E_j^2 + B_j^2}; \ F_{j-1}^2 = \frac{D_j^1 - A_j^1 \cdot F_j^1}{A_j^1 E_j^1 + B_j^1}.$$

The model contains $n$ computational nodes, with $n-2$ finite difference equations. There are two supplementary equations required to solve the system, the boundary conditions, depending on the flow regime.

The supercritical flow regime necessitates the specification of two additional boundary conditions at the upstream boundary. The sub-critical regime requires one upstream and one downstream boundary condition. When the uniform flow is reached, the flow is considered to have a normal depth.

The boundary condition obtained from the water level calculations should be placed far away from the area of interest to ensure the accuracy of the results. Assuming that the initial conditions are the known discharges $Q_j$, and water level $h_j$ for a gradual flow rate, the computational procedure begins from the downstream boundary conditions, where the water level $h_L$ is maintained constant.

This condition may be satisfied if $E_{j+1}^2 = 0$ and $F_{j+1}^2 = h_L$, where j is the total number of $x$-grid points. From Equation (14) there are determined $E_{j-2}^2, F_{j-2}^2, E_{j-3}^2, F_{j-3}^2$, etc. Finally, $E_1^2, F_1^2$ may be determined as the first sweep. Since $Q_1^{m+1}$ is known for the upstream by the flood hydrograph, $h_2^{m+1}$ may be determined downstream, as the second sweep.

## 3. Results

The standard approaches are based on separate evaluations of the maximum flow rates and the volumes of transported sediments, corresponding to the probability P(%). In reality, the maximum flow rate and sediment volume cannot both have the same probability for a flood.

To solve this problem, two extreme scenarios of interest for water management and environmental protection are analyzed, based on floods recorded in the last 20 years:

- Maximum flow rate, using the Bistrita River as an example, and minimum volume of transported sediments in the uncertainty interval;
- Maximum volume of transported sediments based on the hypothesis of flood hydrographs with lower flow rates but longer duration, using the Siret tributaries. Table 5 shows the results for the uncertainty intervals for maximum flood discharge and entrained/deposited sediment volume up to a probability, P = 10%.

**Table 5.** Uncertainty interval for the flow rate and transported sediments.

| P (%) | Gamma | GEV | Frechet | LogPearson | $Q_{inf}^{max}$ | $Q_{sup}^{max}$ | Gamma | Weibull | LogPearson | InvGaussian | $V_{min}$ | $V_{max}$ |
|---|---|---|---|---|---|---|---|---|---|---|---|---|
| 0.1 | 5937 | 7157 | 7449 | 6293 | 5937 | 7449 | 2468 | 2245 | 2468 | 2608 | 2245 | 2608 |
| 0.5 | 4847 | 5372 | 5481 | 5073 | 4846 | 5800 | 2004 | 1879 | 2013 | 2070 | 1879 | 2070 |
| 1 | 4363 | 4676 | 4734 | 4535 | 4050 | 4734 | 1799 | 1712 | 1809 | 1840 | 1580 | 1750 |
| 3 | 3578 | 3651 | 3656 | 3667 | 3400 | 3850 | 1468 | 1430 | 1477 | 1477 | 1400 | 1477 |
| 5 | 3200 | 3203 | 3194 | 3255 | 3030 | 3255 | 1310 | 1290 | 1317 | 1308 | 1230 | 1317 |
| 10 | 2668 | 2617 | 2597 | 2685 | 2597 | 2960 | 1089 | 1088 | 1094 | 1078 | 1088 | 1094 |
| 20 | 2104 | 2045 | 2024 | 2094 | 2024 | 2104 | 856 | 866 | 860 | 844 | 860 | 870 |
| 25 | 1912 | 1859 | 1840 | 1897 | 1840 | 1912 | 778 | 789 | 781 | 767 | 760 | 790 |
| 30 | 1750 | 1706 | 1689 | 1732 | 1689 | 1750 | 712 | 723 | 715 | 703 | 701 | 723 |
| 40 | 1480 | 1455 | 1443 | 1463 | 1443 | 1480 | 603 | 612 | 607 | 598 | 582 | 612 |
| 50 | 1254 | 1249 | 1243 | 1240 | 1240 | 1254 | 512 | 519 | 517 | 512 | 511 | 519 |
| 60 | 1053 | 1065 | 1065 | 1046 | 1046 | 1066 | 433 | 435 | 437 | 436 | 433 | 440 |
| 70 | 864 | 889 | 896 | 866 | 864 | 896 | 359 | 357 | 363 | 366 | 357 | 366 |
| 75 | 770 | 800 | 811 | 778 | 770 | 811 | 323 | 319 | 327 | 331 | 319 | 332 |
| 80 | 674 | 707 | 723 | 687 | 674 | 723 | 286 | 281 | 290 | 295 | 281 | 295 |
| 90 | 462 | 488 | 516 | 491 | 462 | 516 | 207 | 200 | 208 | 215 | 200 | 215 |
| 95 | 328 | 332 | 369 | 367 | 328 | 369 | 158 | 153 | 156 | 163 | 153 | 163 |
| 97 | 259 | 240 | 283 | 302 | 240 | 302 | 134 | 131 | 129 | 133 | 129 | 134 |

Flood hydrographs on large rivers, such as the Siret, do not simply record an increase in flow rate followed by a decrease. In reality, there is a maximum followed, preceded by, or both, by other high values produced by the floods from tributary rivers.

Generally, there is a time gap between the floods on the tributaries and their overlapping with the increased flows on the main course. In our case the maximum value of the hydrograph is obtained when the flow rates from tributaries overlapped with the flow rate of the Siret River.

Figure 6a depicts the discretization of the analyzed domain, while Figure 6b,c illustrate examples of sediment transport possibilities during the flood. The flow rate was calculated using equal time steps from the $Qi$—flood hydrograph.

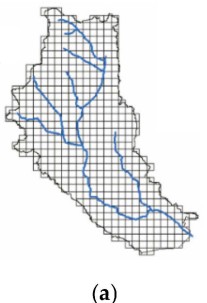

(a)

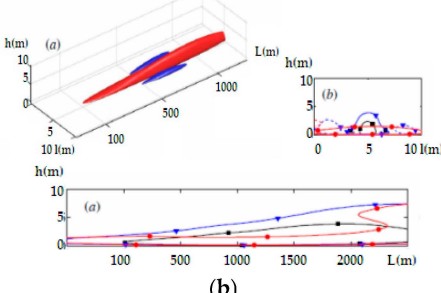

(b)

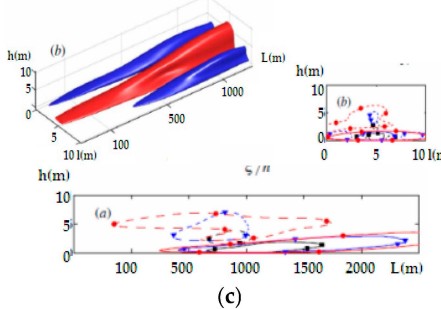

(c)

**Figure 6.** Numerical modeling of the sediment transport: (**a**) Discretization of the interest area; (**b**) Sediment transport-initial phase; (**c**) Sediment transport during the flood.

The transported volume—$V_t$—is typically recorded during the increasing period of the flood hydrograph, while the sediment deposits—$V_d$—are typically recorded when the flow rate decreases. When the hydrographs show multiple flood peaks or when successive floods are recorded at short intervals, as in 2020, new problems arise.

The average multi-annual flow rate is 250 m³/s for the Siret River, with historical minimum $Q_{min}$ = 45 m³/s and maximum $Q_{max}$ = 4650 m³/s. The transported solid sediments have an average velocity of 95 kg/s and a volume of 5.98 million tons per year. It contains about 10% dragged alluvium. Coarse alluvial deposits have the greatest thickness in the Marasesti-Doaga area, reaching over 100 m, and decreasing to about 40 m in the Jorasti-Vulturu area and 15–20 m in the Milcov-Bordeasca area to the south.

In two scenarios, numerical modeling was carried out. The first occurred when the flood from the Bistrita River merged with the existing flood on Siret. The highest flow rate recorded on Bistrita in recent decades was 353 m³/s. At the Frumosu station, a historical flow rate of $Q_{max}$ = 772 m³/s was recorded in 2020. The second scenario assumes a longer but less intense flood. Figure 7 illustrates the area chosen for modeling.

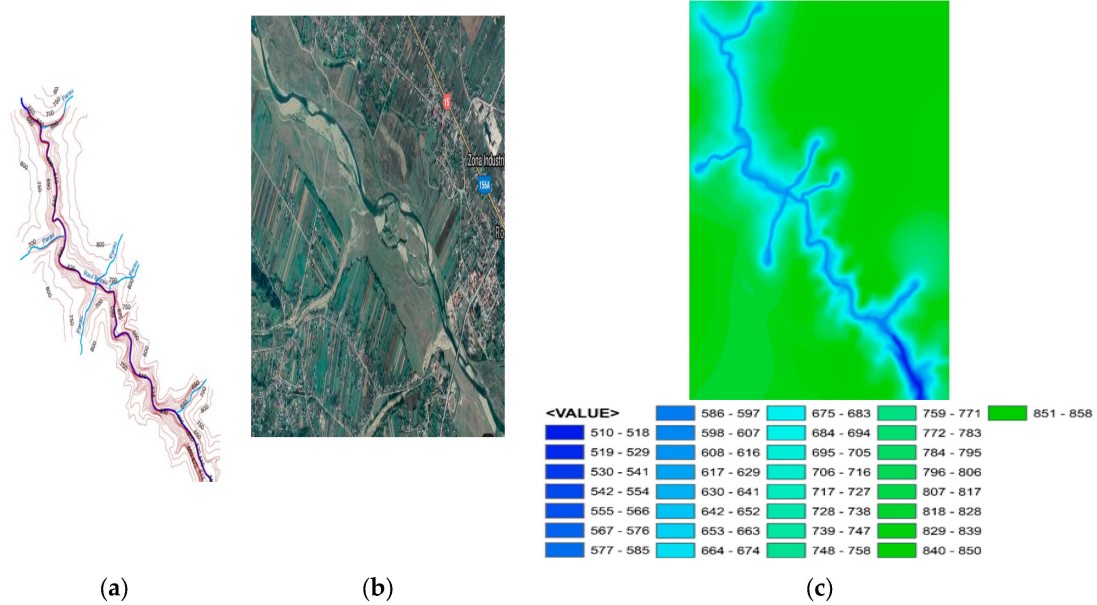

**Figure 7.** Selected area: (**a**) Domain; (**b**) Google Earth view; (**c**) Data for numerical modeling.

Scenario 1 shows erosion in all of the selected cross-sections, with large, flooded areas. Scenario 2 features alternating deposition and erosion sections. Figure 8 presents the possibility of sediment transport; deposited sediments are red, entrained sediments are blue, and recirculated sediments are yellow. The cross-section axis is denoted by 0 and the dimensionless riverbed width is denoted by ±1.5.

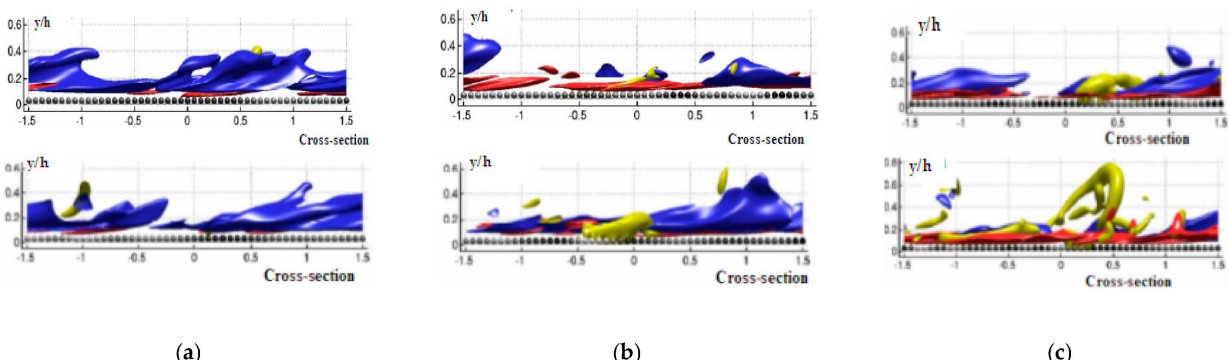

**Figure 8.** Images during modeling the sediment transport phenomena: (**a**) Small flow rates, only few sediment transported; (**b**) High flow rates entrained and deposited sediments; (**c**) Very high flow rates, rapid changes in the riverbed.

Selected parameters: the Coriolis coefficient $\alpha = 1.1$, three values for the constant erosion $C' = 0.043$, $C' = 0.06$ and $C' = 0.084$, depth variation $h_0 = 1$–8 m, cross-sections dimensions, hydraulic conductivity $K = 10$–300 m/day along the watershed, with average values $K = 30$–100 m/day in the north part and average transmissivity $\theta = 100$–500 m$^2$/day with higher values $\theta = 1000$–3000 m$^2$/day in the Focsani area.

The mathematical model used creates the required balance of input and output for each analyzed section. Input data:

- Initial cross-sections: $S_1$: length $l_1 = 70$ m, $S_2$: $l_2 = 100$ m, $S_3$: $l_3 = 140$ m, $S_4$: $l_4 = 200$ m;
- Distances between these sections: $d_1 = 1200$ m, $d_2 = 2350$ m, $d_3 = 3500$ m, $d_4 = 1750$ m;
- River slope: $0.5^o/_{oo}$;
- The average annual rainfall is 500–600 mm, with 450 mm in the summer.

### 3.1. The Risk Zones

To simulate natural flow and sediment transport, the physics of the phenomena must be understood, as well as the significant parameters and their relationships. Different values of sediment roughness were chosen for each cross-section. Figure 9 illustrates some flood propagation scenarios in cross-sections on lateral walls. The initial riverbed sediments are shown in red, and the sediment level at various time intervals is shown in green dashed lines. Figure 10 shows sediment transport in two cross-sections, section 2 and section 4, at various time steps, during periods of increased and decreased flow rate from the flood hydrograph.

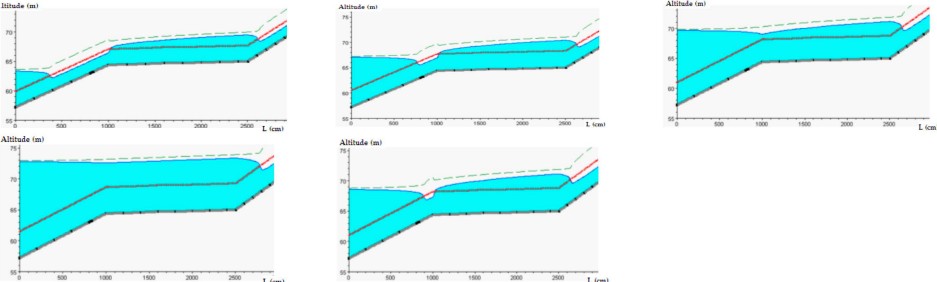

**Figure 9.** Images during the flood modeling showing increase and decrease of the flow rate.

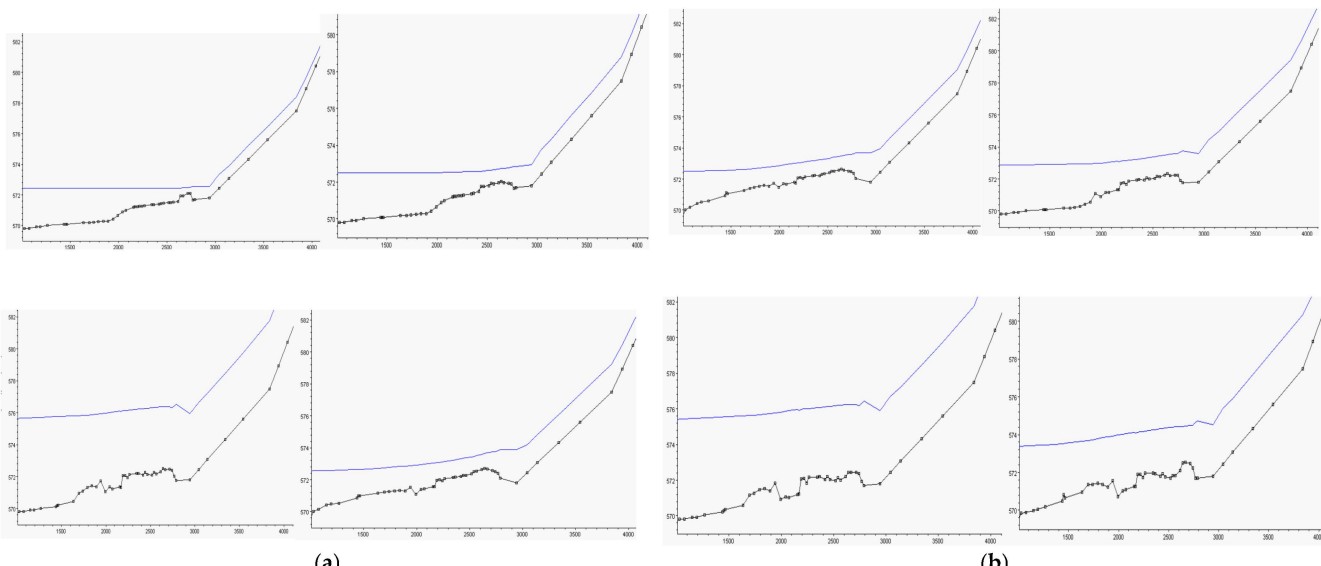

(**a**)  (**b**)

**Figure 10.** Sediment transport during floods: (**a**) section 2; (**b**) section 4.

The calculation scheme described above was tested in a case study with a longitudinal bed slope of 0.005–0.0001. The Manning roughness coefficient is $n = 0.0167$ on the bottom of the riverbed and $n = 0.045$ on the side banks; 9072 grid points with different length and width steps were chosen.

To illustrate, the flow rate transport was selected from the interesting area of five cross-sections where floods were previously recorded in large surfaces. A mass balance is achieved between the entered and transported sediments from one section to the next one. Figure 11 depicts these cross-sections during the flood, starting with the beginning of the hydrograph. The free level of water at the maximum flow rate is marked with a dark blue line; it can be seen in cross-sections where there are lateral spills.

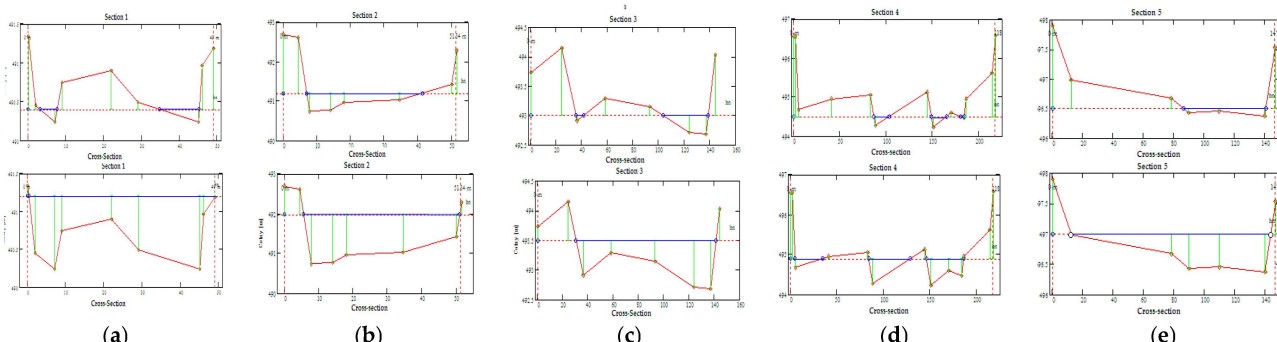

**Figure 11.** Cross-sections and water level at the initial and peak flow rate: (**a**–**e**) sections 1–5.

Changes in the alluvial riverbed are possible, particularly in sections 3–5. For each interval of the flood hydrograph, the program was run for 50 time steps of Δt = 7200 s.

Finally, Figure 12 shows the flooded surfaces in 2020 after the first and second floods overlapped. Yellow represents the first flood, and green represents the second. The flooded communities are highlighted in red.

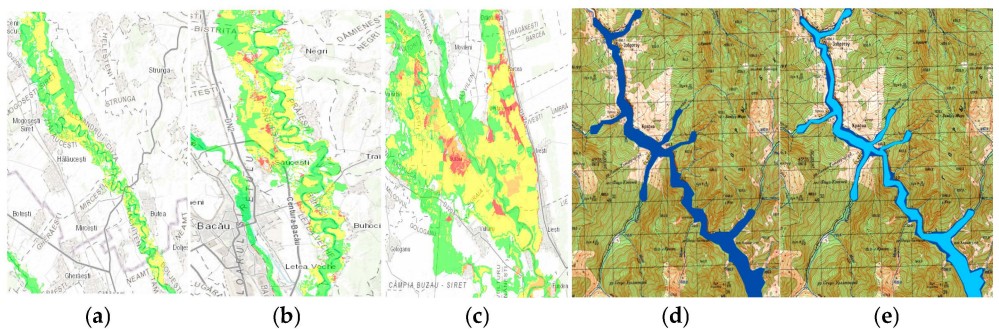

**Figure 12.** Numerical results for the flooded surfaces: (**a**) Covered surfaces after the first flood; (**b**) After the second flood; (**c**) after the third flood; (**d**) Covered surfaces after the third flood; (**e**) Comparison between the first and the third flood.

During the floods from the Bistrita River, the mass balance of solid sediments obtained was $G_i$ = 37.4 kg/s, with $C_g$ = 19.8 kg/s of suspended sediments; the rest was transported. After 52 h, the flow rate in the Panagarati area was $Q$ = 2280 m$^3$/s, $Q$ = 1720 m$^3$/s in Vaduri, and $Q$ = 2325 m$^3$/s in Reconstructia.

There are differences between the covered surfaces with water for the two floods. Although the values of the maximum flow rate are close, the first flood could not be properly evacuated in time (there were still flooded areas) and because of this, the second overlapped and flooded larger surfaces.

Based on numerical modeling, field observations, and data collected after the floods of the last two decades, the risk areas were determined. Table 6 shows the most affected zones and distances at risk of flooding.

The transit of flood waves of 2212 m$^3$/s without forecast implies the discharge of a flow of 1480 m$^3$/s on Siret. If high flow rates from downstream Pangarati tributaries are superposed, the discharge capacity at all other dams is exceeded and the flood wave cannot be transited. Lake Pangarati no longer has an attenuation volume; however, it has a small, useful volume so it can no longer mitigate the high flows from the captured basin. The flood flows are discharged by opening the spillways at a lifting speed of 0.4 m/s. Its initial utile volume of approximately 6.4 million m$^3$ is today only 2.1 million m$^3$. Under these conditions, the evacuation capacity of the Pangarati Lake is exhausted. The Vaduri Lake has a relatively small volume and cannot mitigate the high flows from the basin captured by this lake either. The util volume of the initial lake of 5.60 million m$^3$ is now

only 2.39 million m$^3$ due to massive silting. Lake Reconstructia was supposed to have a regularizing effect on the upstream accumulation. Since the upstream lakes only allow the passage of flood flows, it can no longer take over from the volume of the flood either. Its initial useful volume of 5.6 million m$^3$ is also only 2.4 million m$^3$ today. They currently operate under difficult conditions. It is impossible to capture large amounts of water from floods due to their massive siltation (some over 70%, e.g., Pangarati and Vaduri). The floods cannot be stopped. Additionally, human errors in the exploitation of these lakes in 2005 produced the amplification of flood waves by superimposing the discharged flows above the naturally maximum-formed flood. According to recent analysis, most hydropower lakes in Romania have only 20–40% utile water storage capacity for water supply and electricity production. This means that the additional flows can no longer be stored and floods move downstream, inundating large areas where the transverse riverbeds are not high enough to transport the excess water.

**Table 6.** The risk zones.

| River | Siret | Putna | Bistrita | Bistrita | Cuejd | Cracau | Racaciuni | Trotus | Oituz |
|---|---|---|---|---|---|---|---|---|---|
| Place | Movileni | Putna | Lunca | Costisa | Garceni | Magazia | Racaciuni | Faget | Oituz |
| **Distance (km)** | 21 | 7 | 32 | 22 | 10 | 14 | 11 | 5 | 11 |
| River | Dragomirna | Bistrita | Slanic | Siret | Tazlau | Siret | Siret | Siret | Siret |
| Place | Mitoc | Piatra Neamt | Slanic Moldova | Oituz | Tazlau | Solont | Caiut | Milcov | Ramnicu Sarat |
| Distance (km) | 42 | 82 | 20 | 25 | 72 | 13 | 5 | 63 | 120 |

Floods with effluent flows of 1400–1500 m$^3$/s can only be discharged at the limit if the downstream lakes have been emptied beforehand; moreover, in the area of the confluence between Bistrita and Siret the flow rate is below 600 m$^3$/s. Evacuation of these flows must be less than 6 h, otherwise the flood wave will become uncontrollable. A flood greater than 2835 m$^3$/s cannot be discharged under any circumstances without endangering downstream structures and communities unless more than 12 h of flood mitigation is possible and special consideration is given to discharging the flows from other dams.

*3.2. Proposed Solutions*

Sediment transport has increased dramatically due to a lack of protection dikes, torrential erosion, and poor flood management. In time, massive sediment transport produced rapid clogging of hydropower lakes and the apparition of islands covered with vegetation at the end of the lakes. The transit capacity of excess water from floods was drastically reduced.

To minimize the effects of the flood as much as possible, flood management is proposed, which includes the construction of temporary dams, dikes, and bank slopes, as well as the creation of buffer zones in which some quantities of water can be maintained and used later, as shown in Figure 13a–d.

However, the best solution remains unclogging the reservoirs and hydropower lakes from the Bistrita River and restoring the minor bed of the main tributaries and the Siret, especially the upper part. Under these conditions, large volumes of water can be stored in lakes to be used later, but higher flow rates can also be transported until the discharge into the Danube.

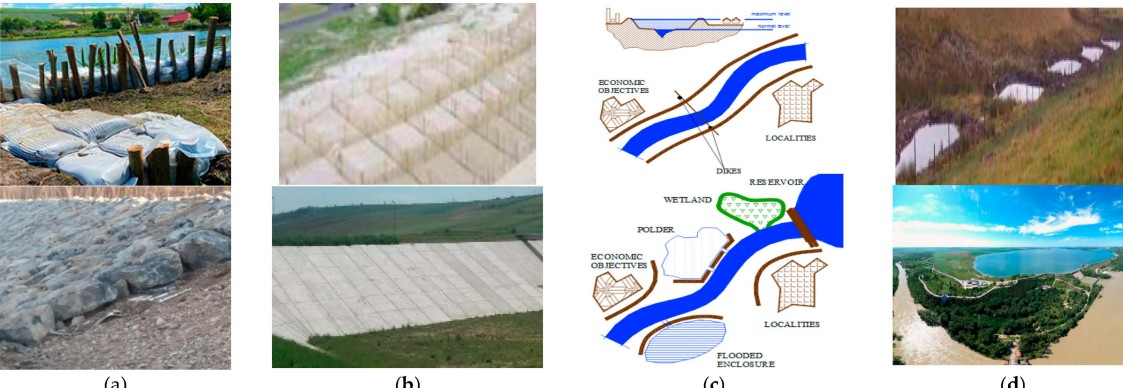

**Figure 13.** Management plans to transient the floods in Siret basin: (**a**) Temporary dams; (**b**) Lateral bank slopes; (**c**) different proposed solutions; (**d**) Buffer zones, reservoirs.

## 4. Discussion

Until 4–5 years ago, the average floodable area in the Siret basin was considered 105,500 ha, with 108 affected localities and a population of 90,902 people exposed to the risk of flooding. However, after the 2020 floods, which were considered exceptional, the flooded area was 293,595 ha, with 39,753 houses destroyed. Unfortunately, in the last 20 years, an annual average of 48,160 ha of flooded surfaces were recorded, approximately 22% of which were represented by agricultural lands.

Climate change affects more and more countries with totally different geographical conditions. In some countries, the melting of glaciers is observed, such as in countries from the northern part of Europe and America. In other parts of the world, very high temperatures have been recorded in the last decade (Spain, France, Portugal, e.g.).

In Asia, climate change produced high and long-lasting monsoon rains. In 2017, millions of people in Bangladesh, India, and Nepal were affected by floods caused by monsoon rains. In northern and central Bangladesh, 3.9 million people were directly affected by floods, and in India, in four states in the north—Assam, Bihar, Uttar Pradesh, and West Bengal—more than 18 million people were also affected. The Cherrapunji and Mawsynram stations recorded the highest rainfall since 1940. In 2022, according to the Pakistani government, which declared a state of emergency, more than 33 million people were affected by floods when water covered a third of the country.

In Romania, floods are primarily caused by the clogging of the existing hydropower lakes on the Bistrita River downstream of Piatra Neamt. No hydropower lake can take over from the tributary flows; they can only transit the water. According to recent analysis, many hydropower lakes have only 20–40% utile volume storage capacity for water supply and electricity production; the rest is clogged. This means that the additional flows can no longer be stored. Floods move downstream and inundate large areas, where the transverse riverbeds are not high enough to transport the excess water. Another cause of floods in this area is caused by significant rainfall, well above the national average. For example, the Trotuș River caused a flood of 3270 $m^3/s$ in 1991, and Bistrita of 4680 $m^3/s$ in 2020. The Trotuș River, together with its tributary, Tazlau, has the highest recorded maximum specific discharge in Romania, of 16 $m^3/s$ per $km^2$. In total, 5 flow rates higher than 1500 $m^3/s$ have been recorded in the last 20 years on this river, corresponding to a probability of 5%. The Belci dam was destroyed by the flood of 1991, and even today it is not restored. As a consequence, the Siret riverbed was massively affected.

When abundant precipitation is recorded because there are destroyed and unrepaired dams and clogged hydropower lakes, large surfaces and many localities are flooded. In 2020, three large floods were recorded in less than two months. It was practically impossible to evacuate the high volume of water from one flood and the next one arrived, causing even more damage.

Numerical modeling of surfaces that can be flooded, regardless of the conditions that produce the floods, is of great help to the local authorities. Knowing the risk areas and identifying the local problems permit finding solutions to protect the agricultural lands and population in the area.

It is easier to mathematically and numerically model a flood and to know its effects than to live the respective experience and draw subsequent conclusions. Any flood, even a small one, causes economic and material damage. The population in the respective area is affected by the lack of drinking water; may experience destroyed houses and flooded agricultural land, which may even lead to a lack of food in certain areas with a dense population; but, most importantly, various transmissible diseases may also appear in these areas.

## 5. Conclusions

Supplementary factors that affect the proper management of floods may be mentioned:

- Excessive deforestation in large areas of watercourses reception basins;
- Inadequate works, such as bridges, footbridges and dams, location of houses, or economic objectives—such as sawmills, gravel and sand mining stations, wood and reed processing—that occupy the minor riverbed;
- Household annexes and fences in major riverbeds, as well as the storage of household waste, sawdust, and wood. During floods, household garbage or wood left on illegally deforested slopes is transported and accumulated in meandering sectors or at narrow bridge openings;
- Under-sizing of bridges and decks.

Reduced transit capacity of flows reduces life safety. It is critical to find solutions to these inconveniences. The risk management plan for floods must be updated at least every six years.

**Funding:** This research received funds from SC Hidroelectrica SA, Research Contract 814/2020.

**Data Availability Statement:** Not applicable.

**Acknowledgments:** The author thanks SC Hidroelectrica SA for letting her participate in two campaigns of monitoring the environmental parameters and collecting samples of sediments from the Siret hydrographic basin. The author also had access to data registered on riverbeds and average tributary flow rates in normal and flood conditions.

**Conflicts of Interest:** The author declares no conflict of interest.

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
