# Peer review of "Mathematical and Numerical Modeling of Repeated Floods from the Siret Basin, Romania, a Risk for Population, Environment, and Agriculture"

_water, doi:10.3390/w15061103_

Round 1
Reviewer 1 Report
Repeated floods, risk for the local population, environment, and agriculture; mathematical and numerical modeling
It focuses on repeated floods, risk for the local population, environment, and agriculture, mathematical and numerical modeling. The manuscript in its current form is not adequate to be published. Authors are requested to revise it and follow the following suggestions to improve their work. In my opinion, some details on the experiment should be presented in methodology that validates your results with filed data. This study looks very localized needs to add some literature of other countries or region which have faced same problem sand make it more readable for globally.
Other comments
Revise your abstract.
* What are the key results?
* What are the practical implications of your research (how can the results be utilized by e.g., readers, community)?
Line 8 What is mean of complex management plan?
Line 19-20 Finally, the main causes of these repeated floods on the Siret River, the zones at per manent risk of flooding, and some solutions to mitigate their effects, protect the population, agri-cultural lands, and the environment, are examined.
On what basis you add this statement in abstract
Introduction
Introduction is poorly framed. Very localized introduction! I suggest the authors to revise it completely into a new format as suggested papers. The authors need to clarify the novelty in the introduction section. This section is not clearly explaining overall picture of your paper. It is suggested to add some literature globally and make it more attractive for readers. Specific issue of your study area and finally problem formulation and scope of the research.
I assume that any change in the introduction section is not necessary, but one of the important tasks after publishing a study is to increase its chance to be seen by the most possible number of researchers. So, the more you cite similar publication, the more the chance that the search engine in the publisher website propose you paper to the researcher. Besides of that, it will also complete your introduction section. As another advantage, it rises new ideas to the researchers by combining various methods or resolving drawback of one seen paper by reading the similar one or extending the methodology to a fully automatic one. So, based on these points, I would like to ask to cite to the similar latest publication. I suggest following paper for your guideline.
Manzoor, Z.; Ehsan, M.; Khan, M. B.; Manzoor, A.; Akhter, M. M.; Sohail, M. T.; Hussain, A.; Shafi, A.; Abu-Alam, T.; Abioui, M. Floods and flood management and its socio-economic impact on Pakistan: A review of the empirical literature. Frontiers in Environmental Science 2022, 10, 1-14. https://doi.org/10.3389/fenvs.2022.1021862
Dawood, F., Akhtar, M. M., & Ehsan, M. (2021). Evaluating urbanization impact on stressed aquifer of Quetta Valley, Pakistan. DESALINATION AND WATER TREATMENT, 222, 103-113.
Sohail, M. T.; Hussan, A.; Ehsan, M.; Al-Ansari, N.; Akhter, M. M.; Manzoor, Z.; Elbeltagi, A. Groundwater budgeting of Nari and Gaj formations and groundwater mapping of Karachi, Pakistan. Applied Water Science 2022, 12, 1-24. DOI: 10.1007/s13201-022-01795-0.
Line 59-69 It is suggested to properly provide the reference of this paragraph.
2. Materials and Methods
Make a clear research motivation and why you think it is necessary to carry on this work. The research gaps need to be clearly articulated.
This section looks like report. It is suggested to rewrite it in proper journal format.
Line 142 Table 1. Average flow rates, Q [m3/s]. what is source of information of this table
Please provide the refences of all equations has been used in this paper
The current methodology is very long in length. It is suggested to remove unnecessary information and write it in concise form.
3.2. Risk zones and proposed solutions
What you proposed not clear
4. Discussion
This section needs to revise and cite most recent studies which support your results.
Line 514 According to numerical modeling,
What it mean, I think it may be like this numerical modeling results indicates …..
What are your suggestions to improve the disaster management and quick response systems in your study area?
What do you propose to cope with the upcoming global warming and its aftershocks? (It has been stated that Pakistan's contribution to greenhouse gasses is less than 1% globally, but it is one of the top ten countries that is affected by global warming (floods, natural disasters, increase in the sea water level).
It is a serious issue of debate. I expect some appreciable and serious contribution from this platform. I advise you to revise it accordingly and provide some useful insights that are helpful to your study area and the international community.
Authors should make a comparative analysis of regional countries and present the effective solution that is vital to cater to such situations.
Author Response
Dear Reviewer,
Thank you for your comments. I hope that now the paper is clearer.
Next, I mention the corrections I made, marked in yellow:
- I rewrote the Abstract of the paper;
- I substantially modified the Introduction:
- I mentioned the novelty of the paper, its purpose, and objectives.
- I added the mentioned bibliographic references and supplementary many more.
- I pointed out where similar problems are encountered
- I corrected lines 8, 19-20, 59-69, 142, and 514
- In the chapter Materials and Methods I showed how this paper can be used in other areas facing similar problems
- I shortened the Materials and methods chapter; I added references for the formulas used
and I explained all the used terms
- I modified in Results the previous paragraph 3.2; Now it is divided into 3.2 - Risk zones and 3.3 - Proposed solutions
- Thanks to you, I understood what catastrophic floods mean. I read what happened in Pakistan last year (and generally, in Asia, in the last five years). More people were affected there than the entire population of Romania. In our case, however, the problem is that flooded areas appear in the Siret hydrographic basin almost every year (but obviously, on a different scale).
I think that the way it is structured now, the paper will be useful to those who deal with similar problems in flood management.
I hope that the paper is now clearer, easier to read and understand, and can be published.
Thank you,

Reviewer 2 Report
Repeated floods, risk for the local population, environment, and agriculture; mathematical and numerical modeling
Interesting paper to read with. However, there are some issues in structuring of this paper. Therefore, I want the author to re-structure the paper.
For example; what you have in Lines 25 – 37 are more to Study area section. You have started your paper with all geophysical details of the river and river basin. That is okay for a report. However, not for a research paper.
You should start the paper in a broader way and explain the research gap. Then you find the solution for the research gap and present it in the paper.
You have very good content in the paper; however, very poor in presentation.
Figure 2 and 3 have photographs. Who have captured them? If it is by the author, please give it.
Please re-arrange your manuscript and I would like to read it and give my technical inputs.
1. Please explain the equations in depth and their physical means.
2. Please verify your results in depth.
3. Please include what others have done in the same thread in other parts of the world.
Author Response
Dear Reviewer,
Thank you for your comments. I hope that now the paper is clearer.
Next, I mention the corrections I made, marked in yellow:
- I rewrote the Abstract of the paper;
- I substantially modified the Introduction:
- I mentioned the novelty of the paper, its purpose, and objectives.
- I added new bibliographic references.
- I pointed out where similar problems are encountered
- I corrected all mentioned lines.
- In the chapter Materials and Methods I showed how this paper can be used in other areas facing similar problems
- I shortened the Materials and methods chapter; I added references for the used formulas
and I explained all the terms for the sediment transport
- For Figure 2 and Figure 3, with photos, I mentioned who made them and I supplemented the references
- I modified the Results chapter; Now it is divided into 3.2 - Risk zones and 3.3 - Proposed solutions
- I mentioned some other places of the world, confronted with massive floods
I think that the way it is structured now the paper will be useful to those who deal with similar problems in flood management.
I hope that the paper is now clearer, easier to read and understand, and can be published.
Thank you,

Round 2
Reviewer 1 Report
I agree with the revision.
Reviewer 2 Report
I think, authors have improved their manuscript and enhanced the quality. Therefore, I would like to accept their effort.